# The Effect of SGLT2 Inhibitors on the Development of Contrast-Induced Nephropathy in Diabetic Patients with Non-ST Segment Elevation Myocardial Infarction

**DOI:** 10.3390/medicina59030505

**Published:** 2023-03-04

**Authors:** Uğur Özkan, Muhammet Gürdoğan

**Affiliations:** Department of Cardiology, School of Medicine, Trakya University, Edirne 22030, Turkey

**Keywords:** contrast induced nephropathy, coronary angiography, percutaneous coronary interventions, SGLT2 inhibiors

## Abstract

*Background and Objectives:* Percutaneous procedures using contrast agents are modern diagnosis and treatment methods for cardiovascular diseases. Contrast use may cause nephropathy, especially in diabetic patients. SGLT2 inhibitors have strong cardioprotective and renal protective effects. In our study, we investigated the effectiveness of this drug group in preventing the development of Contrast-Induced Nephropathy (CIN). *Materials and Methods:* The results of 312 diabetic patients who underwent CAG were analyzed. The study group included 104 DM patients using SGLT2 and the control group did not use SGLT2. These groups were compared with each other in terms of clinical, demographic, and laboratory parameters. *Results:* The groups were similar characteristics. However, post-CAG creatinine values compared with before the procedure, the development of CIN was observed to be significantly less in the group using SGLT2 inhibitor (*p* = 0.03). When the results of the multivariate analysis were examined, it was seen that the use of SGLT2 inhibitors significantly reduced the risk of CIN (odds ratio (OR): 0.41, 95% confidence interval (CI): 0,142–0.966, *p* = 0.004). *Conclusions:* Our study showed that SGLT2 inhibitors may be protective against the development of CIN, especially in patients with comorbid conditions such as diabetes.

## 1. Introduction

Cardiovascular diseases are considered to be the leading cause of morbidity and mortality worldwide [1]. Coronary angiography (CAG) and percutaneous coronary interventions (PCI) are diagnosis and treatment methods using contrast agents that are frequently used in the diagnosis and treatment of cardiovascular diseases. The use of contrast media is considered safe for nephropathy in patients with normal renal function at baseline or in the absence of underlying co-morbid conditions such as advanced age, heart failure, hemodynamic instability, and diabetes mellitus [2]. Acute renal failure secondary to contrast agent use is reported as the 3rd most common cause of acute renal failure in the inpatient population after ischemic nephropathy and toxic nephropathy (secondary to medication) [3]. Diabetes is accepted as a major risk factor for cardiovascular diseases and even it is emphasized that the probability of CIN development in elective coronary procedures of diabetic patients is up to 21.5% [3,4]. The accepted mechanisms in the literature about the pathophysiology of CIN development show that contrast agents cause nephropathy in the kidney both through ischemia and with direct toxic effects [5,6,7,8]. Sodium-glucose cotransporter-2 (SGLT2) inhibitors are a group of cardioprotective drugs that have been shown in large clinical studies beyond their antidiabetic efficacy [9,10]. Glycosuria, natriuresis, and diuresis are well-known clinical effects of SGLT2 inhibitors [11]. However, it has been reported that SGLT2 inhibitors regulate energy homeostasis in the nephron by affecting intraglomerular hemodynamics, show pleiotropic effects with their anti-inflammatory and antifibrotic properties, and consequently reduce glomerulosclerosis and tubulointerstitial fibrosis [12,13]. Studies have demonstrated that SGLT2 inhibitors can reduce the risk of progression of diabetic kidney disease, slow the decline in estimated glomerular filtration rate (eGFR) and decrease urinary albumin excretion. The nephroprotective effects of SGLT2 inhibitors may be attributed to their ability to reduce intraglomerular pressure, improve renal oxygenation, and decrease inflammation and fibrosis in the kidney. Overall, SGLT2 inhibitors offer a promising therapeutic option for patients with Type 2 diabetes and chronic kidney disease, providing both glycemic control and nephroprotection [14].

The pleiotropic effects of SGLT2 inhibitors on kidney function may provide a potential benefit in reducing or even preventing contrast-induced toxicity. In this study, it was aimed to investigate whether there is a relationship between SGLT2 inhibitor use and the development of CIN in diabetic patients who underwent coronary angiography and/or percutaneous coronary intervention.

## 2. Method

### 2.1. Study Population

This was a retrospective study in a single-center and included 104 patients who had a history of Type 2 DM, used SGLT2, and underwent CAG and/or PCI between 1 January 2020 and 1 July 2022. In the control group of the study, 208 DM patients who underwent CAG and/or PCI and did not use SGLT2 in the same period were included. Table 1 and Table 2 show the demographic characteristics and laboratory findings of the study populations.

Patients who used SGLT2 inhibitors in the DM treatment regimen before CAG and/or PCI were included in the study. The study for patient standardization was performed only with patients who underwent CAG with the diagnosis of NST-ACS. Patients with active infection (including COVID-19), patients with congestive heart failure (EF < 40%), patients with malignancy, patients with autoimmune disease, patients using chronic anti-inflammatory drugs, patients with severe anemia, patients with eGFR, estimated Glomerular Filtration Rate (eGFR) < 30 mL/min/1.73 m^2^, patients who used nephrotoxic agents, patients with cardiogenic shock, and patients with a previous history of CIN were excluded from the study to ensure study standardization.

Our study was approved by the Trakya University Medical Faculty Ethics Committee (TUTF-BAEK 2022/1) and complied with the Helsinki Declaration.

### 2.2. Clinical Data Collection

Demographic characteristics, medical histories, laboratory results, and echocardiographic and angiographic data of the patients included in the study (312 patients) were accessed and recorded via the hospital automation system. The values of serum creatinine levels of the participating patients measured before the procedure (CAG and/or PCI) and 48–72 h after the procedure were recorded. The diagnosis of CIN was determined as a 0.5 mg/dL (absolute) or 25% (relative) increase in creatinine value within 48 h, an increase in creatinine level of more than 1.5 times the baseline within 7 days, or a urinary output of less than 0.5 mL/kg/h for at least 6 h after using the contrast agent compared to its level before the procedure [3]. The formula that was used to calculate the eGFR value was: GFR (mL/min/1.73 m^2^) = 175 × (Scr) − 1.154 × (Age) − 0.203 × (0.742 if female) [3].

### 2.3. Coronary Angiography, Percutaneous Intervention, and Medications

CAG and/or PCI indications have been established within the framework of current cardiology guidelines as a result of history, ECG, and cardiac biomarkers [15,16,17,18]. CAG and PCI procedures were performed by expert cardiologists using standard femoral/radial routes. Ejection fraction (LVEF) values (Vivid7; GE Medical Systems, Horten, Norway) obtained from the recorded echocardiography reports of the patients as a result of volumetric evaluation with the modified Simpson’s method before the procedure were recorded [19].

### 2.4. Statistical Analysis

The normality test of the data was performed using the Shapiro–Wilk test. Descriptive data were given as mean ± standard deviation (normally distributed) and median ± (minimum-maximum) (non-normally distributed). Independent sample t-test (normally distributed) and Mann–Whitney U test (non-normally distributed) were used to compare continuous data. Categorical data were compared using the Chi-square test. The effects of the variables alone on the development of CIN were examined by univariate analysis. The effects of the variables on the development of CIN were examined with multivariate analysis. A *p*-value of <0.05 was accepted as significant in the analyses. Binary logistic regression analysis was used for univariate and multivariate analysis. All data were analyzed with SPSS version 21.0 for Windows (SPSS Inc., Chicago, IL, USA).

## 3. Results

Among the 312 participants included in the final analysis, the median age was 59.5 years (45–80) and 189 (60.6%) were male. The demographic characteristics and medical histories of the patients are shown in Table 1. There was no difference between the two groups in terms of demographic characteristics including age, gender, and medical history (*p* > 0.05, for all). Medication rates such as ACE inhibitors, ARB inhibitors, Beta blockers, Calcium channel blockers, diuretics, etc., used before the procedure due to their medical history were similar in both groups and were not statistically significant. The laboratory findings of the patients included in the study before and after CAG and/or PCI are shown in Table 2. When laboratory findings before CAG and/or PCI are examined, it is seen that there is no statistically significant difference between the two groups (*p* > 0.05). However, when the creatinine values measured after CAG and/or PCI were compared with the values measured before the procedure, the development of CIN was observed to be significantly less in the group using an SGLT2 inhibitor (*p* = 0.03). While the rate of CIN development was found as 13.5% in the diabetic patient group using an SGLT2 inhibitor, it was found to be 30.8% in the group who did not use it. There was only one patient in the non-SGLT2 group who required dialysis at follow-up. The results of univariate and multivariate analyses evaluating the risk factors affecting the development of CIN are shown in Table 3. When the results of univariate logistic regression analysis were examined, it was determined that the use of SGLT2 inhibitor, eGFR value, and high EF value were protective in terms of the development of CIN. However, advanced age, smoking, and previous history of coronary revascularization have been shown to increase the risk of CIN. It is also seen that the risk of developing CIN increases with the prolongation of the time spent as a diabetic and the increase in the amount of contrast agent exposure (*p* < 0.05, for all).

When the results of the multivariate analysis were examined, it was seen that the use of SGLT2 inhibitor (odds ratio (OR): 0.41, 95% confidence interval (CI): 0.142–0.966, *p* = 0.004) and the high eGFR value (odds ratio (OR): 0.722, 95% confidence interval (CI): 0.591–0.881, *p* = 0.001) were found to be the only protective factors for CIN. On the other hand, it was observed that the use of insulin, which was insignificant before, became significant (*p* = 0.043). In the multivariate analysis, the most significant effect triggering the development of CIN was the duration of DM (odds ratio (OR): 3.791, 95% confidence interval (CI): 1.716–8.374, *p* = 0.001) followed by low eGFR levels. Other measures such as EF were not found to be significant in the multivariate analysis and are indicated as non-significant (ns) in the table.

## 4. Discussion

The most important finding obtained as a result of this study: The risk of nephropathy due to the use of contrast media after CAG and/or PCI procedures in diabetic patients with SGLT2 inhibitor in the treatment regimen was found to be significantly lower than in the patient group who did not use SGLT2 inhibitors. This is one of the pioneering studies in the literature showing that SGLT2 inhibitors may have a potential benefit in reducing or preventing the development of CIN and is, to our knowledge, the first study with an isolated NST-ACS patient population [20,21,22].

Epidemiological data indicate that the prevalence of Type 2 DM has increased in parallel with the increasing prevalence of obesity all over the world [23,24]. It is underlined that diabetes is accepted as a major risk factor for cardiovascular diseases and even the presence of diabetes should be considered as the equivalent of coronary artery disease [25]. Therefore, coronary procedures for both diagnostic and therapeutic purposes are frequently performed in diabetic patients. It has been reported that the risk of nephropathy secondary to contrast agent use is 4–5 times higher in diabetic patients compared to the normal population [3,26]. It has been shown in previous studies that this risk is especially higher in emergency PCI procedures where renal protection cannot be performed, and even reaches 60% to 90% [27,28]. In our study, the rate of CIN development in the diabetic group who did not use SGLT2 inhibitors was found to be 30.8%, which was significantly higher in line with the literature data. In our study, the rate of development of CIN was found to be 13.5% in the diabetic patient group using SGLT2 inhibitors. It is seen that there is a statistically significant difference between the two groups. In both groups, the volume of contrast agents to which the patients were exposed and the number of cases of acute coronary syndrome was similar to each other.

The mechanisms most emphasized in the literature about the pathophysiology of CIN development can be listed as follows: (1) medullary hypoxia secondary to renal vasoconstriction, (2) endothelial dysfunction associated with increased oxidative stress, (3) direct toxic effect that occurs in the first 24 h after the use of contrast agent and causes mitochondrial dysfunction, desquamation, and the formation of occlusive tubular structures in the proximal tubule cells, (4) the formation of a pro-inflammatory environment and, (5) nephrotoxicity caused by reactive oxygen radicals [5,26,28,29].

Immediately after the use of CM, transient renovascular vasodilation occurs, followed by prolonged vasoconstriction. In our study, the fact that smoking was a significant factor in increasing the development of CIN is consistent with the fact that smoking is a potent vasoconstrictor [5]. As the duration of diabetes increases, vascular smooth muscle cell functions and endothelial functions deteriorate. It has been shown that this condition causes atherosclerosis, impaired nitric oxide-mediated vasodilation, and inflammatory cell migration [29]. In our study, the correlation between the increase in the time spent as a diabetic and the development of CIN is consistent with the literature.

In a study by Zager et al., it was shown that exposure to contrast media causes deterioration in mitochondria functions in renal proximal tubules and disrupts the energy balance in nephrons [29]. SGLT2 inhibitors are a group of drugs that inhibit glucose reuptake by affecting the proximal tubules, mainly in diabetic patients, and perform diuresis and natriuresis. In addition, this group of drugs also blocks the secondary active transport pathway. In secondary active transport, reabsorption takes place at a rate of 1:1 by the sodium-glucose cotransporter pump. Therefore, the high-capacity transporter SGLT2 pathway is responsible for the majority of renal glucose reabsorption in the proximal tubule. This pathway is one of the pathways that cause the nephron to need the most energy after the primary transport of sodium in diabetic patients. SGLT2 inhibitors block this pathway, reducing the energy requirement of the nephron [30,31,32]. One of the reasons why the development of CIN was significantly lower in patients using SGLT2 inhibitors in our study may be that it reduces the energy needed in the renal system and makes energy use more efficient.

In a study by McCullough et al., the direct cytotoxic effect of contrast agents on the basement membrane was demonstrated. It has been shown that this condition, which has been shown to cause desquamation and luminal obstruction in tubular cells, leads to an increase in intratubular pressure and a decrease in GFR [33]. In another study by Mehran et al., it was emphasized that contrast agents create a hyperosmotic environment in the renal tubules and therefore it may be beneficial to dilute the contrast agent and remove it, and for this purpose, the effect of intravenous furosemide was investigated [34,35,36,37]. Again, there are studies in the literature showing that theophylline may be beneficial in preventing the development of CIN due to its diuretic effect [7]. The lower incidence of CIN in patients with high GFR who are exposed to contrast agents is also explained by this mechanism. The fact that the risk of developing CIN decreased as the eGFR value increased in our study is consistent with the literature data. In light of these data in the literature, the strong diuresis effect of SGLT2 inhibitors can be interpreted as a potential reason for the development of less CIN in the group of diabetic patients undergoing CAG and/or PCI and using SGLT2 inhibitors.

In the literature, it is stated that there is a strong correlation between the increase in systemic and vascular inflammation and the development of CIN [33,35]. In fact, it has been reported in clinical studies that inflammatory molecules such as IL-1β, TNF-α, and IL-6 play a role in the development of CIN [38,39,40,41,42]. In similar studies, it has been shown that insulin resistance develops over time in Type 2 diabetic patients and this situation has a proinflammatory effect [41]. As a result of insulin resistance developing in this patient group, the amount of endogenously produced insulin begins to be insufficient and exogenous insulin is added to the treatment regimen. In other words, insulin therapy in these patients is correlated with increased insulin resistance and an underlying proinflammatory state [43,44]. In the multivariate analysis of our study, the probability of developing CIN was found to be higher in the group using insulin. In a study by Kim et al., it was shown that SGLT2 inhibitors suppress the secretion of IL-1β and other proinflammatory cytokines in diabetic patients [45]. Another potential mechanism for the significantly lower development of CIN in diabetic patients using SGLT2 inhibitors in our study may be the release of proinflammatory cytokines from this drug group.

The development of CIN is a complication that increases the length of hospital stay, mortality, and morbidity rates despite successful coronary intervention procedures. Although many factors have been described in the literature that predicts the development of CIN after CAG and/or PCI, widely accepted treatment options for the prevention of CIN are insufficient [28,34]. Preprocedural and post-procedural IV hydration therapy and high-dose statin therapy before the procedure are the most effective methods in CIN prophylaxis. In addition, there are limited data in the literature showing that amlodipine, theophylline, and phosphodiesterase-5 inhibitors may be beneficial [7]. SGLT2 inhibitors, on the other hand, are a group of drugs that are a part of the current treatment of diabetic patients and whose cardioprotective and nephroprotective effects have been strongly accepted in light of current studies. SGLT2 inhibitors may be useful for nephroprotection in patients with elective coronary intervention who will be exposed to contrast media and in diabetic patients who are predicted to be at high risk of developing CIN due to comorbidities.

The main limitations of our study can be listed as follows. First of all, our study was conducted in a single center and with a retrospective design. Although a definition based on the change in creatinine value, which is generally accepted in the literature, was used for the definition of CIN in our study, another limitation is that more specific laboratory markers of contrast-induced nephropathy such as neutrophil gelatinase-associated lipocalin (NGAL) and Cystatin C were not evaluated [46]. The fact that the effect of SGLT2 inhibitors on the development of CIN was investigated only in the NSTEMI patient group in our study can be considered a limitation.

## 5. Conclusions

Diabetic patients with an SGLT2 inhibitor in their treatment regimen have a significantly lower risk of nephropathy associated with the use of contrast media than patients who do not use this drug group. The potent pleiotropic effects of SGLT2 inhibitors may be protective or preventive against the development of contrast-induced nephropathy. In light of future multicenter studies, this may make SGLT2 inhibitors a part of therapy for the patient population scheduled for elective PCI or at high risk of developing CIN.

## Figures and Tables

**Table 1 medicina-59-00505-t001:** Demographic Characteristics of the Study Populations.

Characteristics	No Previous SGLT2i	Previous SGLT2i	*p*-Value
	(*n* = 208)	(*n* = 104)	
Age (years)	58.2 (52–71)	60 (52–70)	0.49
Male gender, *n* (%)	123 (59.1)	66 (63.4)	0.54
Hypertension, *n* (%)	180 (86.5)	88 (84.6)	
ACE inh.	122 (67.7)	60 (68.1)	0.78
ARB inh.	45 (25)	20 (22.7)	
CCB	42 (23.3)	22 (25)	
Diüretic	52 (28.8)	25 (28.4)	
β Blocker	20 (11.1)	10 (11.3)	
Smoking, *n* (%)	56 (26.9)	26 (25)	0.82
Drinking, *n* (%)	40 (19.2)	14 (13.5)	0.42
Prior revascularization	88 (42.3)	52 (50)	0.27
ASA	79 (38)	49 (47.1)	0.07
Statin	103 (49.5)	60 (57.7)	0.10
Fibrate	16 (7.7)	7 (6.7)	0.47
Prior metformin Rx (%)	148 (71.2)	80 (76.9)	0.50
Prior insulin treatment	80 (38.5)	46 (44.2)	0.55
Diabetes Mellitus duration	5.5 ± 2.6	5.4 ± 2.1	0.73

Abbreviations: SGLT2i, Sodium-glucose cotransporter-2 inhibitors.

**Table 2 medicina-59-00505-t002:** Laboratory Findings and Procedural Characteristics of the Study Populations.

Characteristics	No Previous SGLT2i	Previous SGLT2i	*p*-Value
	(*n* = 208)	(*n* = 104)	
Glucose, (mg/dL)	128 (77–223)	122.5 (56–174)	0.07
HbA1c%, (mmol/L)	6.7 ± 0.5	6.6 ± 0.4	0.11
Left ventricular ejection fraction (%)	55.5 (42–72)	54 (42–64)	0.19
Creatinine preprocedural, (mg/dL)	1.28 ± 0.32	1.3 ± 0.3	0.13
Creatinine 3-day postprocedural, (mg/dL)	1.46 (0.74–2.15)	1.39 (0.64–2.08)	**0.035**
Creatinine 7-day postprocedural, (mg/dL)	1.42 (0.7–2.09)	1.3 (0.6–2.17)	**0.046**
eGFR (preprocedural (MDRD, mL/min/1.73 m^2^)	56 (31–116)	51 (31–115)	0.28
Hb preprocedural, (g/dL)	11.8 ± 1.7	11.8 ± 1.2	0.94
Contrast volüme, (mL)	170 (70–290)	170 (70–350)	0.34
Number of implanted stents	1 (0–3)	1 (0–3)	0.96
HDL, (mg/dl)	38 (25–60)	33 (24–62)	0.50
LDL, (mg/dl)	91 (50–159)	99.5 (50–177)	0.65
Triglyceride, (mg/dl)	135.5 (98–231)	154 (36–264)	0.62
Total cholesterol, (mg/dl)	177 ± 48.3	166 ± 43.8	0.22
ALT, (U/L)	19.5± 3.9	20.8± 4.9	0.13
AST, (U/L)	17.6± 3.2	18.7± 3.1	0.06
Contrast-induced nephropathy n, (%)	64 (30.8)	14 (13.5)	**0.03**
Dialysis requirement *n*, (%)	1	0	
PCI procedure *n*, (%)	176 (84.6)	90 (86.5)	0.78

Abbreviations: ALT, alanine aminotransferase; AST, aspartate aminotransferase; β, Beta; CCB, calcium chanal blocker; eGFR, estimated Glomerular Filtration Rate; Hemoglobin A1c, HbA1c; HDL-C, high-density lipoprotein cholesterol; LDL-C, low-density lipoprotein cholesterol; PCI, percutaneous coronary intervention. Bold results are <0.05, statistically significant.

**Table 3 medicina-59-00505-t003:** Univariate and Multivariate Predictors of Contrast-induced Nephropathy in Patients with Diabetes Mellitus.

	Univariate Analysis *			Multivariate Analysis **		
	OR	95% CI	*p*	OR	95% CI	*p*
SGLT2 use	0.35	0.13–0.942	**0.038**	0.41	0.142–0.966	**0.004**
Age	1.118	1.05–1.191	**0.001**	-	-	ns
Gender	0.806	0.315–2.060	0.652	1.938	1.293–2.634	**0.016**
Hypertension	2.007	0.419–9.62	0.383	-	-	ns
Smoking	4.8	1.78–12.941	**0.002**	2.43	2.164–2.729	**0.01**
Prior revascularization	2.727	1.038–7.164	**0.042**	-	-	ns
Prior metformin	2.807	0.762–10.339	0.121	-	-	ns
Prior insülin treatment	1.404	0.553–3.563	0.476	1.519	1.095–2.109	**0.043**
DM duration	2.583	1.693–3.939	**<0.001**	3.791	1.716–8.374	**0.001**
Drinking	0.718	0.187–2.75	0.629	-	-	ns
Glucose	0.998	0.985–1.012	0.795	-	-	ns
HbA1c	1.306	0.477–3.575	0.603	-	-	ns
LVEF%	0.923	0.862–0.989	**0.023**	-	-	ns
eGFR preprocedural mL/min/1.73 m^2^	0.906	0.86–0.954	**<0.001**	0.722	0.591–0.881	**0.001**
Hemoglobin	0.85	0.618–1.17	0.319	-	-	ns
Contrast volume	1.014	1.003–1.025	**0.01**	-	-	ns

Cox and Snell Square = 0.530; Nagelkerke R square = 0.813; Accuracy = 0.779. * Logistic Regression (Method = Enter), ** Logistic Regression (Method = Backward Stepwise (Wald)). Abbreviations: DM, Diabetes Mellitus; eGFR, estimated Glomerular Filtration Rate = mL/min/1.73 m^2^); Hemoglobin A1c, HbA1c; ns, non-significant; SGLT2, Sodium-glucose cotransporter-2 inhibitors. Bold results are <0.05, statistically significant.

## Data Availability

The data presented in this study are available on request from the corresponding author.

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
