# Peer review of "The Effect of SGLT2 Inhibitors on the Development of Contrast-Induced Nephropathy in Diabetic Patients with Non-ST Segment Elevation Myocardial Infarction"

_medicina, 2023, doi:10.3390/medicina59030505_

Round 1
Reviewer 1 Report
This is a nicely accomplished study of contrast induced nephropathy which concludes that those patients who were treated with SGLT2 inhibitors prior to coronary angiography had a reduced incidence of contrast induced nephropathy. The number of patients is impressive at 312. The conclusion is quite important. However, there are flaws which must be addressed carefully in order to substantiate this finding.
1) Why are the authors limiting the study to non ST-T wave myocardial infarction?? Surely they perform coronary angiogrpahy for other indications. They could certainly carry out a randomized study of SGLT2 inhi bitors in patients undergoing elective angiography.
2) The biggest problem is the lack of documentation of all prior concomitant medication other than metformin. In order to have unbiased contrast groups we must know that the two groups were equally balanced with respect to ACE inhibitors, ARB inhbitors, Beta blockers, Calcium channel blockers and diuretics as well as insulin.
3)
Author Response
Response to Reviewer 1 Comments
Many thanks for your didactic and comprehensive evaluation of the manuscript. Below, you will find our responses:
Point 1. Why are the authors limiting the study to non ST-T wave myocardial infarction?? Surely they perform coronary angiogrpahy for other indications. They could certainly carry out a randomized study of SGLT2 inhi bitors in patients undergoing elective angiography..
Response 1: : As you know, studies in the literature have shown that the clinical presentation of the patient and the indication for coronary angiography have different effects on the development of CIN in patients. In our study, It was performed for patient standardization. As per your suggestion, the relevant sentence has been added to the Study population section.
“The study for patient standardization was performed only with patients who underwent coronary angiography with the diagnosis of NST-ACS.”
Point 2. The biggest problem is the lack of documentation of all prior concomitant medication other than metformin. In order to have unbiased contrast groups we must know that the two groups were equally balanced with respect to ACE inhibitors, ARB inhbitors, Beta blockers, Calcium channel blockers and diuretics as well as insulin.
Response 2: Pre-procedural medication rates were similar in our study. For this reason, we did not specify them separately in the medical history. As per your suggestion, , the relevant sentence has been added to the Results section and showed at Table 1.
“ Medication rates such as ACE inhibitors, ARB inhibitors, Beta blockers, Calcium channel blockers, and diuretics used before the procedure due to their medical history were similar in both groups and were not statistically significant
Dear Rewiewer, as per your suggestion, the article has been revised by a native speaker. We send you the editorial certificate as an additional document.
Proof of English Editing
Date: February 20, 2023
Manuscript Title: The Effect of SGLT2 Inhibitors on the Development of ContrastInduced Nephropathy in Diabetic Patients with Non-ST Segment Elevation Myocardial Infarction
To whom it may concern,
This document ensures that the manuscript mentioned above has been proofread and/or edited by a native speaker. We hereby confirm that the following issues have been corrected: grammar, spelling, punctuation, sentence structure, phrasing and overall style, and any language errors in this article have been rectified.
Yours sincerely,
Disclaimer: Neither the research content nor the authors’ intentions were altered in any way during the editing process. The manuscript should be English-ready for publication; however, the authors have the ability to accept or reject our suggestions and make changes. We are not responsible for any changes made following our proofreading and editing.
Leo Villa Eğitim ve Danışmanlık
AyÅŸecik Sokak TandoÄŸan Apt. No 19 Daire 2 Fulya ÅžiÅŸli Ä°stanbul Mobil: (0530) 776 95 46 / Ofis: (0212) 272 95 63 www.leovilla.com

Reviewer 2 Report
Thank you for the opportunity to review this manuscript.
The paper “The Effect of SGLT2 Inhibitors on the Development of Contrast-Induced Nephropathy in Diabetic Patients With Non-ST Segment Elevation Myocardial Infarction” by Ozkan et Gurdogan is an interesting original research.
My comments:
1. The issue is not completely new but it is a matter of debate (Perazella MA, Juncos LA. Drug-Induced Osmotic Nephropathy: Add SGLT2-Inhibitors to the List? Kidney360. 2021 Dec 21;3(3):550-553. doi: 10.34067/KID.0007882021. PMID: 35582186; PMCID: PMC9034826.).
2. The Authors investigated the effectiveness of this drug group in preventing the development of contrast induced nephropathy among 312 diabetic patients in a case control study. They found a significantly less development of CIN in the group using SGLT2 inhibitor (p=0.03), concluding that SGLT2 inhibitors may be protective for the development of CIN, especially in patients with comorbid conditions such as diabetes.
3. The introduction section is quite exhaustive. I would like the Authors to consider including a paragraph about “nephroprotective” effects of SGLT2is, other than the cardioprotective they described (SGLT2is and Renal Protection: From Biological Mechanisms to Real-World Clinical Benefits doi: 10.3390/ijms22094441. The EMPA-KIDNEY Collaborative Group; doi: 10.1056/NEJMoa2204233).
4. Methods are appropriate and well described. I would better explain the timeline of CIN definition on the basis of GFR or creatinine changes.
5. In the Table 3, why LVEF is not included in multivariate analysis? Please clarify.
6. Please reformulate “this is the one of the first study in the literature to show that SGLT2 inhibitors may have a potential benefit in reducing or preventing the development of CIN” because of the presence of several clinical trial in the current literature (doi: 10.1016/j.phrs.2022.106597; doi: 10.1097/FJC.0000000000001329; doi: 10.3389/fcvm.2022.918167; …).
7. References prepared not in accordance with the guidelines.
Author Response
Response to Reviewer 2 Comments
Many thanks for your didactic and comprehensive evaluation of the manuscript. Below, you will find our responses:
Point 1. The issue is not completely new but it is a matter of debate (Perazella MA, Juncos LA. Drug-Induced Osmotic Nephropathy: Add SGLT2-Inhibitors to the List? Kidney360. 2021 Dec 21;3(3):550-553. doi: 10.34067/KID.0007882021. PMID: 35582186; PMCID: PMC9034826.)
Response 1: Many thanks for your evaluation .
Point 2. The Authors investigated the effectiveness of this drug group in preventing the development of contrast induced nephropathy among 312 diabetic patients in a case control study. They found a significantly less development of CIN in the group using SGLT2 inhibitor (p=0.03), concluding that SGLT2 inhibitors may be protective for the development of CIN, especially in patients with comorbid conditions such as diabetes.
Response 2: Many thanks for your evaluation .
Point 3. The introduction section is quite exhaustive. I would like the Authors to consider including a paragraph about “nephroprotective” effects of SGLT2is, other than the cardioprotective they described (SGLT2is and Renal Protection: From Biological Mechanisms to Real-World Clinical Benefits doi: 10.3390/ijms22094441. The EMPA-KIDNEY Collaborative Group; doi: 10.1056/NEJMoa2204233).
Response 3: As per your suggestion, the paragraph has been added to the introduction section
“Studies have demonstrated that SGLT2 inhibitors can reduce the risk of progression of diabetic kidney disease, slow the decline in estimated glomerular filtration rate (eGFR) and decrease urinary albumin excretion. The nephroprotective effects of SGLT2 inhibitors may be attributed to their ability to reduce intraglomerular pressure, improve renal oxygenation, and decrease inflammation and fibrosis in the kidney. Overall, SGLT2 inhibitors offer a promising therapeutic option for patients with type 2 diabetes and chronic kidney disease, providing both glycemic control and nephroprotection.”
Point 4. Methods are appropriate and well described. I would better explain the timeline of CIN definition on the basis of GFR or creatinine changes.
Response 4: As per your suggestion, the relevant sentence has been revised.
The diagnosis of CIN was determined as a 0.5mg/dL (absolute) or 25% (relative) increase in creatinine value within 48 hours, an increase in creatinine level of more than 1.5 times the baseline within 7 days, or a urinary output of less than 0.5 mL/kg/h for at least 6 hours after using the contrast agent compared to its level before the procedure [3].
Point 5. In the Table 3, why LVEF is not included in multivariate analysis? Please clarify.
Response 5: As per your suggestion, the relevant sentence has been added to the results section
“Other measures such as EF were not found significant in the multivariate analysis and are indicated by the expression non-significant (ns) in the table.”
Point 6. Please reformulate “this is the one of the first study in the literature to show that SGLT2 inhibitors may have a potential benefit in reducing or preventing the development of CIN” because of the presence of several clinical trial in the current literature (doi: 10.1016/j.phrs.2022.106597; doi: 10.1097/FJC.0000000000001329; doi: 10.3389/fcvm.2022.918167; …).
Response 6: As per your suggestion, the relevant sentence has been revised.
“This is one of the pioneering studies in the literature showing that SGLT2 inhibitors may have a potential benefit in reducing or preventing the development of CIN and is, to our knowledge, the first study with an isolated NST-ACS patient population. “
Point 7. References prepared not in accordance with the guidelines.
Response 7: As per your suggestion, the references have been revised.
Dear Rewiewer, as per your suggestion, the article has been revised by a native speaker. We send you the editorial certificate as an additional document.
Proof of English Editing
Date: February 20, 2023
Manuscript Title: The Effect of SGLT2 Inhibitors on the Development of ContrastInduced Nephropathy in Diabetic Patients with Non-ST Segment Elevation Myocardial Infarction
To whom it may concern,
This document ensures that the manuscript mentioned above has been proofread and/or edited by a native speaker. We hereby confirm that the following issues have been corrected: grammar, spelling, punctuation, sentence structure, phrasing and overall style, and any language errors in this article have been rectified.
Yours sincerely,
Disclaimer: Neither the research content nor the authors’ intentions were altered in any way during the editing process. The manuscript should be English-ready for publication; however, the authors have the ability to accept or reject our suggestions and make changes. We are not responsible for any changes made following our proofreading and editing.
Leo Villa Eğitim ve Danışmanlık
AyÅŸecik Sokak TandoÄŸan Apt. No 19 Daire 2 Fulya ÅžiÅŸli Ä°stanbul Mobil: (0530) 776 95 46 / Ofis: (0212) 272 95 63 www.leovilla.com

Round 2
Reviewer 1 Report
The authors are not realizing my criticisms. This is an enormously interesting observation which must be carefully defended. They must explain why they chose only patients with non ST segment elevation myocardial infarction for their study population. Why not patients receiving catheterization for evaluation of angina? As far as the concomitant medications, that is of extreme importance. They must add a comparison of each major class of cardioactive medications in their SGLT2 and in their control populations.
Author Response
Response to Reviewer 1 Comments
Many thanks for your didactic and comprehensive evaluation of the manuscript. Below, you will find our responses:
Point 1. The authors are not realizing my criticisms. This is an enormously interesting observation which must be carefully defended. They must explain why they chose only patients with non ST segment elevation myocardial infarction for their study population. Why not patients receiving catheterization for evaluation of angina? As far as the concomitant medications, that is of extreme importance. They must add a comparison of each major class of cardioactive medications in their SGLT2 and in their control populations.
Response 1: : First of all, we would like to point out that your suggestions have shed light on us to investigate the effects of SGLT2 inhibitors on the development of CIN in a larger sample group including all subtypes of acute coronary syndromes in the future. It has even been added to the limitations section. However, in this study, we mainly aimed to investigate the effect of SGLT2 inhibitor use on the development of CIN in patients with NSTEMI. There are several reasons why we did not include STEMI patients or patients with low GRACE risk scores in our study population. In STEMI, patients are younger and risk factors are less, clinical presentation is more acute, pathophysiologically prone to systemic inflammation and hypercoagulation is more, however, hemodynamic status is much more variable in STEMI, and left ventricular functions can be suppressed more. Likewise, in patients with low GRACE risk scores, nephrotoxic medications may discontinued 48-72 hours before the procedure, as early angiography is not required, and nephroprotection is performed with iv hydration and other medications in this process, if necessary. All these reasons are reported to be effective on the development of CIN. However, NSTEMI patients exhibit a different behavior in terms of the factors listed above. In conclusion, considering that coronary angiography indications are different clinical conditions, this study was conducted in a more homogeneous patient group to investigate the effect of SGLT2 inhibitors on the development of CIN, as it minimizes the effect of the above-mentioned confounding factors. As per your suggestion, the relevant sentence has been added to the Limitation section.
“The fact that the effect of SGLT2 inhibitors on the development of CIN was investigated only in the NSTEMI patient group in our study can be considered as a limitation. However, this study is valuable in that it paves the way for future investigations of the effects of SGLT2 inhibitors on the development of CIN in larger sample groups including all subtypes of acute coronary syndromes.
In line with your suggestion, the concomitant cardioactive medications was compared and added to Table 1.
